# Over-the-Counter Breast Cancer Classification Using Machine Learning and Patient Registration Records

**DOI:** 10.3390/diagnostics12112826

**Published:** 2022-11-16

**Authors:** Tengku Muhammad Hanis, Nur Intan Raihana Ruhaiyem, Wan Nor Arifin, Juhara Haron, Wan Faiziah Wan Abdul Rahman, Rosni Abdullah, Kamarul Imran Musa

**Affiliations:** 1Department of Community Medicine, School of Medical Sciences, Universiti Sains Malaysia, Kubang Kerian 16150, Kelantan, Malaysia; 2School of Computer Sciences, Universiti Sains Malaysia, Gelugor 11800, Penang, Malaysia; 3Biostatistics and Research Methodology Unit, School of Medical Sciences, Universiti Sains Malaysia, Kubang Kerian 16150, Kelantan, Malaysia; 4Department of Radiology, School of Medical Sciences, Universiti Sains Malaysia, Kubang Kerian 16150, Kelantan, Malaysia; 5Breast Cancer Awareness and Research Unit, Hospital Universiti Sains Malaysia, Kubang Kerian 16150, Kelantan, Malaysia; 6Department of Pathology, School of Medical Sciences, Universiti Sains Malaysia, Kubang Kerian 16150, Kelantan, Malaysia

**Keywords:** Asian women, breast cancer, explainable artificial intelligence, machine learning, medical consultation delays, screening model, clinical decision support systems

## Abstract

This study aims to determine the feasibility of machine learning (ML) and patient registration record to be utilised to develop an over-the-counter (OTC) screening model for breast cancer risk estimation. Data were retrospectively collected from women who came to the Hospital Universiti Sains Malaysia, Malaysia for breast-related problems. Eight ML models were used: k-nearest neighbour (kNN), elastic-net logistic regression, multivariate adaptive regression splines, artificial neural network, partial least square, random forest, support vector machine (SVM), and extreme gradient boosting. Features utilised for the development of the screening models were limited to information in the patient registration form. The final model was evaluated in terms of performance across a mammographic density. Additionally, the feature importance of the final model was assessed using the model agnostic approach. kNN had the highest Youden J index, precision, and PR-AUC, while SVM had the highest F2 score. The kNN model was selected as the final model. The model had a balanced performance in terms of sensitivity, specificity, and PR-AUC across the mammographic density groups. The most important feature was the age at examination. In conclusion, this study showed that ML and patient registration information are feasible to be used as the OTC screening model for breast cancer.

## 1. Introduction

Breast cancer is the most common cancer among women in at least 140 countries [1]. The WHO aims to reduce global breast cancer mortality by 25% annually between 2020 and 2040, which is equivalent to 2.5 million breast cancer death worldwide [2]. Generally, breast cancer affects women above the age of 50 and the risk of having the disease increases with increased age [3,4,5]. The risk factors for breast cancer are mainly divided into two groups [6]. The inherent risk factors include a family history of breast cancer, age, and gender, while the extrinsic risk factors include diet and lifestyle. The risk factors differ according to the individual and population. One of the important risk factors for breast cancer is mammographic density which reflects the amount of dense and fatty tissue in the breast [7,8]. Women with denser breasts had four to six times higher chances of developing breast cancer than those with less dense breasts [9]. Asian women or women with Asian ancestry had denser breasts compared to other populations [10].

Early detection of breast cancer is crucial in reducing the severity of the disease. Any delay in the diagnosis and prognosis may worsen the presentation of the disease. Generally, delay in the management of cancer is divided into two: patient delay and provider delay [11]. Patient delay is the delay during the period between the first discovery of the symptom and medical consultation. Provider delay is the delay between medical consultation and the beginning of the cancer treatment. Additionally, the combination of both types of delay is known as a total delay. However, a more complicated model of the total delay had also been proposed. For example, the total patient delay model detailed the total delay into five stages [12] while the total breast cancer delay model detailed it into eight stages [13]. Nonetheless, a total delay of more than 1 to 3 months has been observed to be associated with advanced stages of cancer and reduced survival of the patients [11,14,15]. Thus, there is a need to improve the efficiency of the medical workflow for breast cancer patients in arranging a medical consultation.

Artificial intelligence (AI) is a subfield of computer science that aims to develop a system capable of performing a task that usually requires human intelligence. The rise of AI is expected to improve many areas including the fields of healthcare and medicine. AI had been studied to be used as a medical analytic tool including for drug discovery, genomic medicine, disease prognosis and diagnosis, and personalised healthcare [16,17]. For example, AI has been shown to aid the diagnosis of fibrotic lung diseases, tuberculosis, and diabetes in research studies [18,19,20]. AI also had been shown to track disease progression in diseases such as systemic sclerosis [21], osteoarthritis [22], and mild cognitive impairment [23], and predict disease complications in diseases such as diabetes [24], Crohn’s disease [25], and atrial fibrillation [26]. However, the adoption of AI in healthcare and medicine is slower than in other fields [27]. Explainable AI (XAI) aims to make the AI more interpretable and understandable to the end-users. Thus, the use of XAI will further help in the successful implementation of AI in healthcare. Generally, the approaches used in the XAI can be divided into model-specific and model-agnostic approaches [28]. Model-specific approaches are limited to specific machine learning (ML) models. One of the main limitations of this approach is that a comparison between models is not appropriate. Model-agnostic approaches overcome this limitation and are applicable to any ML model. XAI had been researched for diagnosis and prediction of glioblastoma [28], colorectal cancer [29], thoracic cancer [30], renal cell carcinoma [31], COVID-19 [32], chronic wounds [33], and Alzheimer’s disease [34]. The use of XAI in medicine is expected to provide insights and transparency into the AI models. Thus, XAI can further help in establishing trust and confidence among medical professionals in the utilisation and implementation of AI in clinical settings [35].

This study aims to develop an over-the-counter (OTC) ML model for breast cancer screening to be deployed in a breast clinic using patient registration records. The model can accelerate the medical workflow for breast cancer management and provide women with a high probability of breast cancer with a timely medical consultation. In other words, women predicted by the model to have a suspicious breast case can be given a high priority for medical consultation with clinicians. Additionally, the performance of the model will be evaluated across dense and non-dense cases. Lastly, we aim to determine the top influential features of the OTC screening model.

## 2. Materials and Methods

### 2.1. Data

Breast cancer data were collected retrospectively from the Breast Cancer Awareness and Research Unit (BestARi), Department of Radiology, and Department of Pathology at Hospital Universiti Sains Malaysia (HUSM). BestARi is a breast cancer resource centre in HUSM. BestARi receives women with breast-related problems from the northeast coast region of Malaysia, especially from the state of Kelantan. The breast cancer data records in BestARi were limited to 1 January 2014 and 30 June 2021. Twenty-seven variables were collected in this study. Twenty-four features were collected from the BestARi including (1) date of examination; eight features related to sociodemographic and personal information: (1) age at examination, (2) race, (3) marital status, (4) number of children, (5) age at menarche, (6) weight, (7) height, (8) handedness; six features regarding the symptoms or patient complaints: (1) lump, (2) nipple discharge, (3) nipple retraction, (4) axillary mass, (5) pain, and (6) skin changes; and nine features regarding the medical history: (1) history of breast surgery or implant, (2) history of breast trauma, (3) history of birth control or hormone replacement therapy, (4) history of the previous mammography, (5) history of breast self-examination, (6) breastfeeding history, (7) history of total abdominal hysterectomy bilateral salpingo-oophorectomy (TAHBSO), (8) family history of breast cancer, and (9) menopausal status. All features were used in the ML model development except for the date of examination as the feature provided no information for the model development. Another two variables collected from the Department of Radiology, HUSM, were breast imaging-reporting and data system (BIRADS) classification information and BIRADS density (or mammographic density). Both variables were used to classify the cases into dense vs. non-dense groups and normal vs. suspicious groups. Finally, the last information collected from the Department of Pathology, HUSM, was histopathological examination (HPE) results. The latter three variables were used to determine the outcome variable. 

The data from the Department of Radiology and Department of Pathology were combined with BestARi’s data if both data were dated within a year after the date of BestARi’s data for each patient. The latest medical record was taken if patients had several records in the BestARi and a single record from the Department of Radiology or Department of Pathology. Afterwards, a body mass index (BMI) was further calculated from the individual weight and height and was added to the existing list of features. Each patient was classified as a normal or suspicious class. The normal class was patients with a BIRADS classification of 1 or who had a diagnosis of normal from the HPE result. The suspicious class was patients with a BIRADS classification of 2, 3, 4, 5, and 6 or who had a diagnosis of benign or malignant subtype of breast cancer from the HPE result. Patients with a BIRADS classification of 0 and missing BIRADS classification or mammographic density were excluded from the study. Additionally, non-dense breast women were those with BIRADS density of A and B, while dense breast women were those with BIRADS density of C and D. Table 1 presents the characteristics of the collected data. 

### 2.2. Pre-Processing Steps 

Initially, all 24 features including the additional variable of BMI were included in the model development. Next, missing values in the data were imputed using a bagged tree model. Subsequently, numerical variables with absolute correlations above 0.8 with other numerical variables were removed. Then, the training dataset was balanced using a random over-sampling examples (ROSE) algorithm [36]. All numerical features were normalised and transformed using a Yeo-Johnson transformation [37]. A dummy coding variable was created for all categorical features for all ML models except for the random forest model. The random forest model had been shown to have at least similar performance if not better when categorical features were used as factor variables as opposed to when the dummy variables were used in the model [38]. The ROSE algorithm was implemented using a **themis** package version 1.0.0 [39]. The remaining pre-processing steps were implemented using a **recipes** package 1.0.1 [40]. 

### 2.3. Machine Learning Models

Eight OTC screening models were developed from ML methods including k-nearest neighbour (kNN), elastic-net logistic regression, multivariate adaptive regression splines (MARS), artificial neural network (ANN), partial least square (PLS), random forest, support vector machine (SVM), and extreme gradient boosting (XGBoost). SVM was implemented using a radial basis function kernel which used a nonlinear class boundary to maximize the width margin between the class. All ML algorithms were implemented using the **parsnip** package version 1.0.1 [41] with the **kknn** package version 1.3.1 [42] as a backend for kNN, **glmnet** package version 4.1-4 [43] for elastic-net logistic regression, **earth** package version 5.3.1 [44] for MARS, **nnet** package version 7.3-17 [45] for ANN, **mixOmics** package version 6.16.3 [46] for PLS, **ranger** package version 0.14.1 [47] for random forest, **kernlab** package version 0.9-31 [48] as a backend for SVM, and **xgboost** package 1.6.0.1 [49] for XGBoost. R version 4.1.3 was used to develop all the screening models [50]. 

### 2.4. Model Comparison and Hyperparameter Tuning

The data were split into 80% development dataset and 20% validation dataset. The development dataset was further split into nested cross-validation groups for model comparison and hyperparameter tuning. The outer folds were split into 10-fold cross-validation groups of 80% training and 20% testing datasets. Each training dataset of each fold was further split into 25 bootstrap samples (inner folds). The validation dataset was further split into a dense breast dataset and a non-dense breast dataset. Thus, there were three validation datasets available: (4.1-41) the whole validation dataset, (2) the dense breast validation dataset, and (4.1-43) the non-dense breast validation dataset.

A random search with a Latin hypercube grid design of 500 combinations of hyperparameters was used for model comparison and hyperparameter tuning. Firstly, all the performance metrics from the results of the bootstrapped samples were summarised by the mean and standard deviation to obtain the descriptive result for each model. The performance metrics of each model were compared using a one-way ANOVA and subsequently pairwise independent *t*-test if the former test was significant. A *p*-value below 0.05 was considered significant. Additionally, the *p*-values for the post hoc pairwise independent *t*-test were adjusted using Bonferroni corrections. Once the best model was identified, the hyperparameters were chosen based on the highest performance metrics from the bootstrapped sample. Figure 1 elucidates the flow of the analysis for this study. Finally, the best model was re-fit using the chosen hyperparameters on the whole development dataset to obtain the final model.

### 2.5. Performance Metrics

Four performance metrics used for model comparison were precision, precision recall-area under the curve (PR-AUC), F2 score, and Youden J index. Once the final model was identified, four hyperparameter tuning results with the highest mean of the aforementioned performance metrics were determined. The best hyperparameters result was selected from the four tuning results based on the highest sensitivity value. The performance metrics were defined below:$$Precision=\frac{TP}{TP+FP}$$
$$Recall/sensitivity=\frac{TP}{TP+FN}$$
$$F2\;score=\left(1+2^{2}\right)\frac{precision\times recall}{2^{2}\times precision+recall}$$
$$Specificity=\frac{TN}{TN+FP}$$
Youden J index=sensitivity+specificity−1

A true positive (*TP*) case was defined as a suspicious case and predicted suspicious by the model, while a true negative (*TN*) case was a normal case and predicted normal by the model. A false negative (*FN*) case was a suspicious case but predicted normal by the model, while a false positive (*FP*) case was a normal case but predicted suspicious by the model. 

### 2.6. Explainable Approach

The model agnostic approach was used to estimate the variable importance for the final ML model. The variable importance was estimated as a mean change in the value of the loss function after variable permutations. The number of permutations was set to 50. The loss function was defined as 1—PR-AUC. The PR-AUC in the loss function reflected the performance of the ML model. Thus, if the feature was important, the performance of the ML model would worsen after permutating the feature. The worse performance of the ML model would in turn result in a high value of the loss function. Hence, the most important feature was the feature with the highest value of 1—PR-AUC. Only the top fifteen important variables were displayed in the variable importance plot. The explainable approach was applied using **DALEX** and **DALEXtra** packages versions 2.4.2 and 2.2.1 [51,52].

## 3. Related Works

Numerous research had been conducted related to breast cancer and ML. Previous studies had used different types of data including imaging modalities, genomic data, and clinical data. Most studies involving ML and breast cancer utilised imaging data especially mammograms and ultrasound [53], while only several studies utilised tabular data. Additionally, a public dataset such as Wisconsin diagnostic breast cancer (WDBC) dataset, despite the tabular nature of the data, the features were derived from the fine needle aspirate imaging of breast mass [54]. Other types of tabular data used for ML classification of breast cancer were sociodemographic, clinical, histological, and pathological data. These types of tabular data were used to predict breast cancer recurrence [55] and survival [56]. Additionally, for breast cancer risk estimation such as screening and diagnosis, imaging data and imaging-derived features were commonly utilised [53]. The use of imaging data in previous studies limited the utilisation of the ML model in the early phase of the screening stage prior to medical consultation.

Several ML algorithms had been used in previous studies that utilised tabular data for the prediction of breast cancer, breast cancer recurrence, and survival of breast cancer patients. Table 2 presents the summary of the previous research related to machine learning classification and breast cancer that utilised tabular data such as sociodemographic, medical history, clinical, pathological, histological, molecular, and genomic data. SVM had been shown to outperform other ML models in several studies involving the prediction of breast cancer recurrence and distant recurrence with the best accuracy at 0.96 [57,58,59]. However, other studies found ANN and random forest had the best performance in predicting breast cancer recurrence [60,61]. Moreover, for the prediction of the survival of breast cancer patients, naïve Bayes, deep learning, and multilayer perceptron (MLP) had the best accuracy at 0.80, 0.83, and 0.88, respectively [60,62,63]. All the aforementioned studies utilised different datasets which may contribute to the difference in the model performance. Additionally, for breast cancer prediction, random forest showed a promising result with accuracy and an area under the curve (AUC) of 0.98 [64]. Other studies showed that XGBoost and MLP had better performance and outperformed random forest in their respective studies [65,66]. However, all three studies except for Hout et al. [66] used clinical data such as the level of glucose, insulin, leptin, and adiponectin which was beyond the initial screening stage of breast cancer. Additionally, a meta-analysis study had shown that SVM outperformed the other classifier such as ANN, decision tree, naive Bayes, and kNN in breast cancer risk estimation [67]. This meta-analysis was limited to ML models performed on imaging data, thus, the performance of the aforementioned ML models as an initial breast cancer screening model utilising a tabular dataset have yet to be explored. 

## 4. Results

### 4.1. Model Comparison

Eight OTC screening models were developed from ML. kNN had the highest Youden J index, precision, and PR-AUC, while the ML model with the highest F2 score was SVM. Table 3 presents the descriptive performance of all ML models, while Figure 2 further illustrates the performance comparison of all models.

One-way ANOVA showed that there was a significant difference between the mean of Youden J index, F2 score, precision, and PR-AUC among all ML models (Table 4). Further post hoc pairwise comparison using *t*-test indicated all pairwise comparisons were significant after Bonferroni correction except for XGBoost vs. elastic-net logistic regression for Youden J index and XGBoost vs. elastic-net logistic regression, ANN vs. elastic-net logistic regression, and XGBoost vs. ANN for F2 score (Figure 3). Thus, kNN was identified to be the best ML model for the purpose of OTC breast cancer screening in this study.

### 4.2. Hyperparameter Tuning

Table 5 presents the four results of hyperparameter tuning with the highest Youden J index, F2 score, precision, and PR-AUC. Models 1, 2, and 4 had lower specificity than sensitivity, while model 3 had it otherwise. kNN model 3 was selected as the best hyperparameters tuning result as it had the highest sensitivity. 

### 4.3. Explainable Approach

Table 6 displays the performance of the final kNN model on the validation dataset across mammographic density. The model had a higher sensitivity on the non-dense cases and a higher specificity on the dense cases. Additionally, the performance differences across the mammographic density were very minimal as shown in Table 5. Furthermore, Figure 4 indicated that there was no difference between PR-AUC of non-dense and dense breast women for the final kNN model as both lines were overlapped. 

Figure 5 illustrates the top fifteen influential features of the final ML model. The top three most influential variables were age at examination, birth control/hormone replacement, and race. In terms of patient complaints, breast pain, breast lump, and breast trauma were the most important factors that influence the model’s prediction as opposed to the other complaints. 

## 5. Discussion

In this study, we evaluated the feasibility of OTC breast cancer screening models developed from ML. The model was aimed to predict women with suspicious breast problems or women with a high probability of developing breast cancer. The screening model used the information obtained during patient registration prior to a medical consultation with the clinician. Thus, patients with a suspicious breast issue would be prioritised at the screening stage and referred to a breast cancer specialist for timely consultation. Previous studies showed that early detection of breast cancer reduces its mortality [68,69]. Additionally, one of the factors of severe breast cancer presentation and poor survival among breast cancer patients was a delay in seeking medical treatment [70,71,72,73]. The development of the OTC screening model would be beneficial in minimising the time between a woman first noticing a symptom and arranging a medical consultation. At least about 17% of women with breast cancer symptoms in European countries had a delayed medical consultation of at least 3 months or more [74]. In southeast Asian countries such as Malaysia, a delay in medical consultation was estimated at 2 months [75]. In general, shortening the delay in arranging medical consultations would be helpful for the prognosis of breast cancer women.

OTC models were developed from eight ML models in this study. The kNN models were significantly better than the other seven models in terms of the Youden J index, precision, and PR-AUC. Additionally, in terms of F2 score SVM had the highest performance value. Thus, the best model based on the four-performance metrics was kNN followed by random forest and ANN. The SVM model had the lowest Youden J index and precision and one of the lowest PR-AUC, despite having the highest F2 score. SVM was believed to work well with imbalanced datasets compared to other ML models, however, this was not the case in our study [76]. Additionally, the final kNN model had a balanced performance between sensitivity and specificity (Table 4). In the hyperparameters tuning stage, we prioritised ML models with a higher sensitivity value. The OTC model aimed to be deployed in the breast clinic during the registration prior to the medical consultation. The model with high sensitivity would prioritise women with a suspicious breast issue which in turn accelerates the needed process for those with medical urgency.

The features used for the development of ML screening models were sociodemographic information, medical history, and patient complaints. A study conducted to develop ML models to predict breast cancer in Chinese women included ten risk factors that achieved the best sensitivity and specificity of 0.66 and 0.69 using XGBoost [66]. This study achieved the sensitivity and specificity of 0.82 and 0.79, respectively, using kNN. Therefore, our study showed that adding patient symptoms or complaints to the features used in the development of the screening model improved the predictive performance of the screening model. Another study conducted to predict breast cancer using laboratory data showed the best precision performance at 0.85 using ANN [65] while the precision for our final kNN model was at 0.81. Although the performance of our model was slightly lower, however, obtaining laboratory data before medical consultation was unfeasible and impractical in our study.

Mammographic density is a known risk factor for developing breast cancer [77]. Asian women had a higher mammographic density than non-Asian women [78,79], thus, having a higher risk of getting breast cancer. For example, in Malaysia, Chinese women had been shown to have denser breasts than the other races [80,81]. A few studies denoted that the proportion of women who attended mammogram procedures in Malaysia was at least half of them were women with dense breasts [82,83]. An ML screening model aimed to be applied to this population should take this information into account. However, it was inappropriate to include the mammographic density as one of the features in the screening model as the density was known at a later stage after medical examination. The final kNN model had a slightly higher sensitivity and specificity in a non-dense and dense group, respectively (Table 6). However, the comparison of the PR-AUC of the model indicated that there was no performance difference between the two groups. Additionally, the explainable ML revealed the most significant feature in the final model was the age at examination. The incidence of breast cancer had been shown to increase with age [84]. However, breast cancer presented at a younger age tended to be more aggressive and at a higher stage of cancer [84,85,86]. Thus, in developing the ML screening model, misclassification of suspicious cases as normal cases especially in younger women could be a catastrophic error. Moreover, there were two modifiable features which were weight and breast self-examination (BSE). Weight control had been suggested to reduce breast cancer risk [87,88]. Although BSE did not relate to breast cancer risk, frequent BSE led to an increased incidence of breast cancer [87]. Additionally, there were three influential features related to patient complaints including breast pain, breast lump and breast trauma.

This study used secondary data collected from a university- and research-based hospital in Kelantan, Malaysia. The data was further validated by a radiologist and pathologist to ensure the good quality of the data. However, our study still had a few limitations. One of the main limitations of this study was the size of the data to develop our screening models. The lack of data was a prevalent issue in the application of ML in healthcare [89]. However, this issue was worsened in our study as the dataset had missing values and imbalanced outcome classification. Subsequently, we used a bagged tree model and ROSE algorithm to overcome these issues, and undeniably larger data will further improve our model. Additionally, we only included one hospital in our study as we utilised information from patient registration records which were specific to the BestARi, HUSM at the time this study was conducted. Including more hospitals in the study was not feasible due to the lack of standardisation in the patient registration record among the hospitals. However, future studies should aim to include more hospitals, if possible, thus increasing the size of the data. Nonetheless, the challenges and approaches presented in the study reflected a real workflow in the development and application of the OTC ML model for breast cancer screening.

## 6. Conclusions

We evaluated eight ML to be developed as an OTC screening model for breast cancer. We used patient registration records including sociodemographic, medical history, and patient complaints as features for the development of the screening models. This study found that the OTC screening models developed from the ML and patient registration records show promising performance. The screening models can be deployed in a breast clinic and improve the workflow of breast cancer management. Thus, the deployment of the model will reduce patient delays in arranging investigations and consultations from the breast cancer team.

## Figures and Tables

**Figure 1 diagnostics-12-02826-f001:**
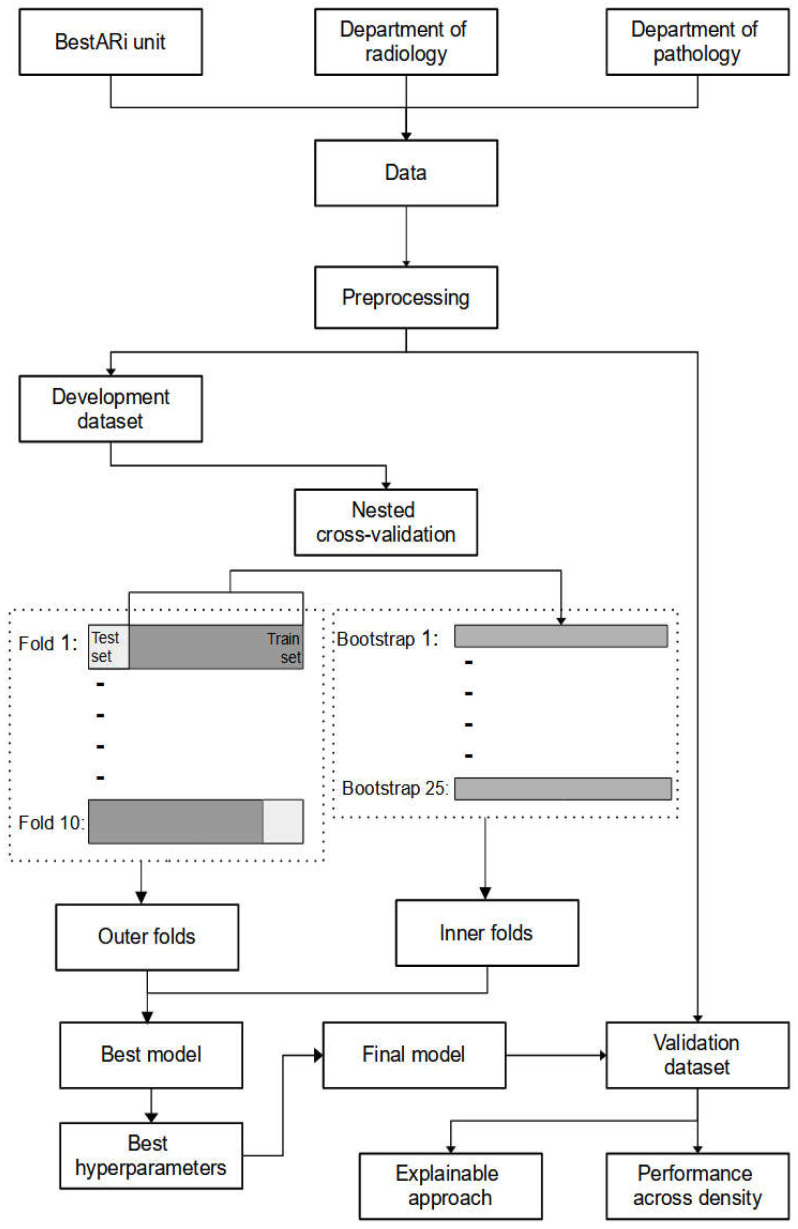
The flow of the analysis.

**Figure 2 diagnostics-12-02826-f002:**
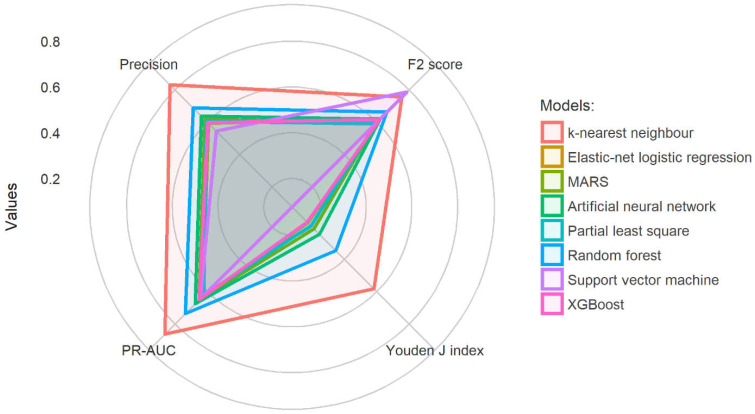
Model comparison across four performance metrics.

**Figure 3 diagnostics-12-02826-f003:**
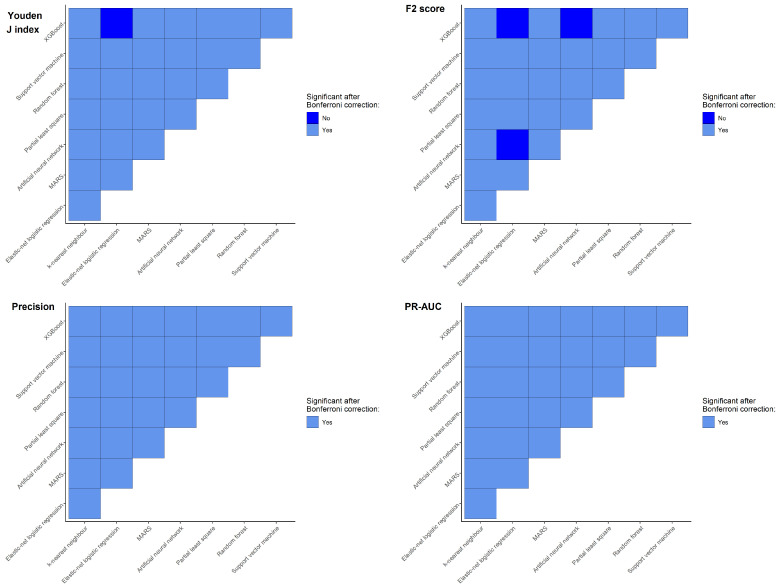
Post hoc pairwise comparison using *t*-test.

**Figure 4 diagnostics-12-02826-f004:**
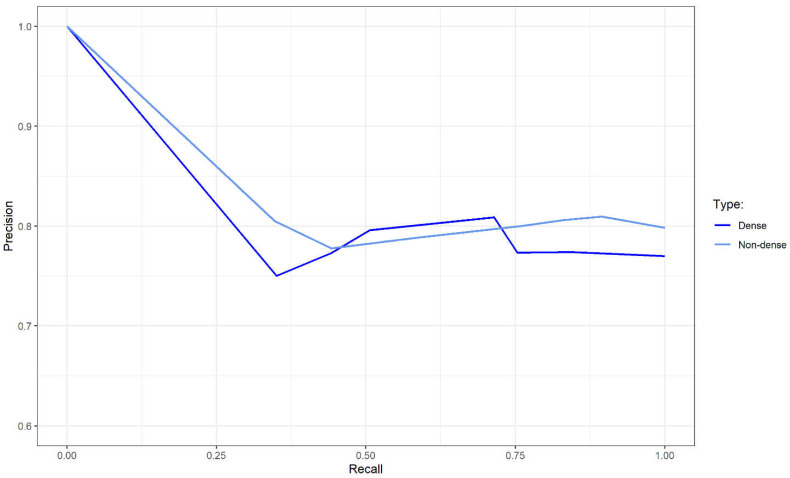
Precision recall-area under the curve for the final machine learning model across mammographic density on the validation dataset.

**Figure 5 diagnostics-12-02826-f005:**
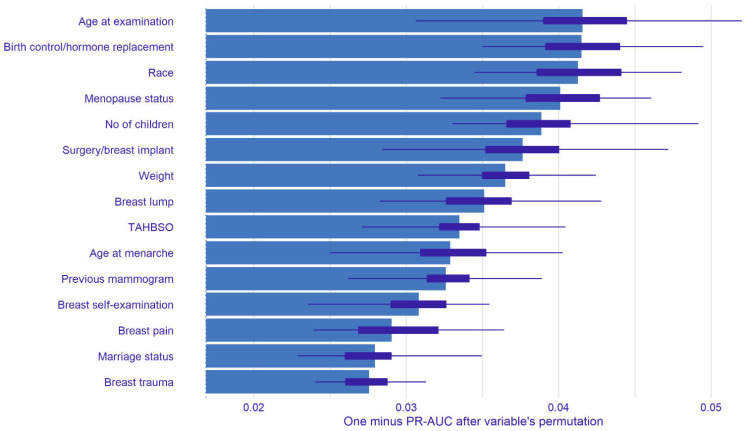
Top fifteen influential features for the k-nearest neighbour model. The bar indicates the mean values of one minus PR-AUC, and the box plot reflects the distribution of the values of one minus PR-AUC.

**Table 1 diagnostics-12-02826-t001:** Characteristics of the features collected from Hospital Universiti Sains Malaysia.

Characteristic	Normal, *n* = 230 *n* (%)	Suspicious, *n* = 861 *n* (%)	Missing Values *n* (%)	Overall, *n* = 1091
Age at examination ^1, 2^	50.0 (8.1)	53.7 (9.6)	3 (0.3%)	52.9 (9.4)
Age at menarche ^1, 2^	13.0 (1.5)	13.1 (1.5)	97 (8.9%)	13.1 (1.5)
No of children ^1, 2^	3.8 (2.7)	3.6 (2.4)	85 (7.8%)	3.7 (2.5)
Weight (kg) ^1, 2^	64.2 (12.9)	63.5 (12.8)	263 (24.0%)	63.7 (12.8)
Height (cm) ^1^	156.4 (5.5)	155.0 (6.4)	692 (63.0%)	155.2 (6.3)
BMI ^1, 2^	27.1 (5.7)	26.7 (5.6)	696 (64.0%)	26.8 (5.6)
Race ^2^			34 (3.1%)	
Chinese	21 (9.4%)	112 (13.4%)		133 (12.6%)
Indian	0 (0.0%)	4 (0.5%)		4 (0.4%)
Malay	201 (89.7%)	706 (84.8%)		907 (85.8%)
Others	0 (0.0%)	3 (0.4%)		3 (0.3%)
Siamese	2 (0.9%)	8 (1.0%)		10 (0.9%)
Marriage status ^2^			59 (5.4%)	
Divorced	0 (0.0%)	4 (0.5%)		4 (0.4%)
Married	208 (95.9%)	759 (93.1%)		967 (93.7%)
Single	8 (3.7%)	46 (5.6%)		54 (5.2%)
Widowed	1 (0.5%)	6 (0.7%)		7 (0.7%)
Breastfeeding ^2^			541 (50.0%)	
No	30 (24.4%)	131 (30.7%)		161 (29.3%)
Yes	93 (75.6%)	296 (69.3%)		389 (70.7%)
Lump^2^			41 (3.8%)	
No	148 (67.0%)	588 (70.9%)		736 (70.1%)
Yes	73 (33.0%)	241 (29.1%)		314 (29.9%)
Nipple discharge ^2^			52 (4.8%)	
No	205 (94.9%)	793 (96.4%)		998 (96.1%)
Yes	11 (5.1%)	30 (3.6%)		41 (3.9%)
Nipple retraction ^2^			45 (4.1%)	
No	213 (97.3%)	784 (94.8%)		997 (95.3%)
Yes	6 (2.7%)	43 (5.2%)		49 (4.7%)
Axillary mass ^2^			55 (5.0%)	
No	203 (94.0%)	764 (93.2%)		967 (93.3%)
Yes	13 (6.0%)	56 (6.8%)		69 (6.7%)
Pain^2^			54 (4.9%)	
No	172 (80.0%)	691 (84.1%)		863 (83.2%)
Yes	43 (20.0%)	131 (15.9%)		174 (16.8%)
Skin changes ^2^			55 (5.0%)	
No	204 (94.0%)	772 (94.3%)		976 (94.2%)
Yes	13 (6.0%)	47 (5.7%)		60 (5.8%)
Breast surgery/implant ^2^			76 (7.0%)	
No	143 (69.1%)	531 (65.7%)		674 (66.4%)
Yes	64 (30.9%)	277 (34.3%)		341 (33.6%)
Trauma ^2^			108 (9.9%)	
No	191 (94.6%)	754 (96.5%)		945 (96.1%)
Yes	11 (5.4%)	27 (3.5%)		38 (3.9%)
BC-HR^2^			51 (4.7%)	
No	130 (59.1%)	554 (67.6%)		684 (65.8%)
Yes	90 (40.9%)	266 (32.4%)		356 (34.2%)
Previous mammogram ^2^			40 (3.7%)	
No	116 (52.5%)	348 (41.9%)		464 (44.1%)
Yes	105 (47.5%)	482 (58.1%)		587 (55.9%)
Breast self-examination ^2^			106 (9.7%)	
No	44 (20.9%)	149 (19.3%)		193 (19.6%)
Yes	167 (79.1%)	625 (80.7%)		792 (80.4%)
Handedness ^2^			667 (61.0%)	
Left	6 (7.4%)	20 (5.8%)		26 (6.1%)
Right	75 (92.6%)	323 (94.2%)		398 (93.9%)
TAHBSO ^2^			70 (6.4%)	
No	187 (86.6%)	720 (89.4%)		907 (88.8%)
Yes	29 (13.4%)	85 (10.6%)		114 (11.2%)
Family history ^2^			520 (48.0%)	
No	101 (80.2%)	352 (79.1%)		453 (79.3%)
Yes	25 (19.8%)	93 (20.9%)		118 (20.7%)
Menopause status ^2^			0 (0.0%)	
No	139 (60.4%)	385 (44.7%)		524 (48.0%)
Yes	91 (39.6%)	476 (55.3%)		567 (52.0%)
Mammographic density			0 (0.0%)	
Non-dense	124 (53.9%)	468 (54.4%)		592 (54.3%)
Dense	106 (46.1%)	393 (45.6%)		499 (45.7%)

Notes: BestARi = breast cancer awareness and research unit; Family history = family history of breast cancer; BC-HR = history of birth control or hormone replacement; TAHBSO = history of total abdominal hysterectomy bilateral salpingo-oophorectomy; ^1^ mean (SD); ^2^ Features included in the model development.

**Table 2 diagnostics-12-02826-t002:** Summary of the previous works related to machine learning classification and breast cancer that utilised tabular data.

Study	Dataset	ML Classifier	Purpose	Performance Metrics ^1^
Kim 2012 [57]	Clinical, histological, and pathological data	SVM ^2^, ANN, Cox regression	Breast cancer recurrence	Accuracy = 0.85 AUC = 0.85 Sensitivity = 0.89 Specificity = 0.73
Ahmad 2013 [58]	Sociodemographic, clinical, and pathological data	DT, SVM ^2^, ANN	Breast cancer recurrence	Accuracy = 0.96 Sensitivity = 0.97 Specificity = 0.95
Cirkovic 2015 [60]	Clinical, histological, and molecular data	ANN ^2^, SVM, LR, DT, NB	Breast cancer recurrence	Accuracy = 0.93 AUC = 0.95 Sensitivity = 0.96 Specificity = 0.83
		ANN, SVM, LR, DT, NB ^2^	Breast cancer survival	Accuracy = 0.80 AUC = 0.83 Sensitivity = 0.65 Specificity = 0.85
Sun 2018 [62]	Clinical and genomic data	DL ^2^, SVM, RF, LR	Breast cancer survival	Accuracy = 0.83 Sensitivity = 0.20 Specificity = 0.95 Precision = 0.75
Kalafi 2019 [63]	Sociodemographic, clinical, and pathological data	MLP ^2^, DT, RF, SVM	Breast cancer survival	Accuracy = 0.88 Sensitivity = 0.96 Specificity = 0.83 Precision = 0.79 F1 score = 0.87
Zeng 2019 [59]	Sociodemographic, clinical, histological, and pathological data	SVM ^2^	Breast cancer distant recurrence	AUC = 0.87 Sensitivity = 0.47 Precision = 0.68 F1 score = 0.56
Hou 2020 [66]	Sociodemographic and medical history	XGBoost ^2^, RF, DL, LR	Breast cancer prediction	Accuracy = 0.67 AUC = 0.74 Sensitivity = 0.66 Specificity = 0.69
Kabiraj 2020 [61]	Sociodemographic and clinical data	RF ^2^, XGBoost	Breast cancer recurrence	Accuracy = 0.75 Sensitivity = 0.94 Specificity = 0.32 Precision = 0.72 F1 score = 0.64
Khatun 2021 [65]	Sociodemographic and clinical data	NB, RF, MLP ^2^, LR	Breast cancer prediction	AUC = 0.89 Sensitivity = 0.85 Precision = 0.85 F1 score = 0.84
Anisha 2021 [64]	Sociodemographic and clinical data	RF ^2^	Breast cancer prediction	Accuracy = 0.98 AUC = 0.98

AUC = area under the curve, SVM = support vector machine, ANN = artificial neural network, DT = decision tree, LR = logistic regression, NB = naive Bayes, DL = deep learning, RF = random forest, MLP = multilayer perceptron. ^1^ Performance metrics of the best or final model in the study. ^2^ Model with best performance metrics/selected as the final model in the study.

**Table 3 diagnostics-12-02826-t003:** Descriptive performance of all machine learning models.

Models	Youden J Index Mean (SD)	F2 Score Mean (SD)	Precision Mean (SD)	PR-AUC Mean (SD)
k-nearest neighbour	0.58 (0.06)	0.75 (0.03)	0.83 (0.04)	0.86 (0.02)
Elastic-net logistic regression	0.17 (0.05)	0.62 (0.06)	0.59 (0.03)	0.63 (0.03)
MARS	0.21 (0.05)	0.60 (0.04)	0.62 (0.02)	0.65 (0.03)
Artificial neural network	0.25 (0.05)	0.62 (0.04)	0.64 (0.03)	0.67 (0.03)
Partial least square	0.19 (0.01)	0.59 (0.01)	0.61 (0.01)	0.62 (0.01)
Random forest	0.35 (0.04)	0.66 (0.03)	0.69 (0.02)	0.74 (0.03)
Support vector machine	0.08 (0.16)	0.79 (0.09)	0.55 (0.08)	0.64 (0.06)
XGBoost	0.17 (0.07)	0.62 (0.09)	0.60 (0.04)	0.65 (0.03)

MARS = multivariate adaptive regression splines, XGBoost = extreme gradient boosting, PR-AUC = precision recall-area under the curve.

**Table 4 diagnostics-12-02826-t004:** Model comparison using one-way ANOVA test.

Models	*n*	Youden J Index		F2 Score		Precision		PR-AUC	
		F-Statistics (df1, df2)	*p*-Value	F-Statistics (df1, df2)	*p*-Value	F-Statistics (df1, df2)	*p*-Value	F-Statistics (df1, df2)	*p*-Value
kNN	5000	21,471 (7, 38,132)	*p* < 0.01	8511 (7, 38,132)	*p* < 0.01	24,768 (7, 38,132)	*p* < 0.01	27,694 (7, 38,132)	*p* < 0.01
EN-LR	5000								
MARS	3140								
ANN	5000								
PLS	5000								
RF	5000								
SVM	5000								
XGBoost	5000								

kNN = k-nearest neighbour, EN-LR = elastic-net logistic regression, MARS= multivariate adaptive regression splines, ANN = artificial neural network, PLS = partial least square, RF = random forest, SVM = support vector machine, XGBoost = extreme gradient boosting, PR-AUC = precision recall-area under the curve.

**Table 5 diagnostics-12-02826-t005:** Top four hyperparameter tuning results of k-nearest neighbour with the highest Youden J index, F2 score, precision, and precision recall-area under the curve.

Model	Fold	Neighbours	Distance Weighting Function	Minkowski Distance	Sensitivity Mean (SD)	Specificity Mean (SD)	Youden J Index Mean (SD)	F2 Score Mean (SD)	Precision Mean (SD)	PR-AUC Mean (SD)
1	1	1	Inversion	1.24	0.77 (0.03)	0.88 (0.03)	0.65 (0.04)	0.78 (0.03)	0.87 (0.03)	0.88 (0.02)
2	3	10	Triweight	1.92	0.76 (0.03)	0.87 (0.03)	0.63 (0.03)	0.78 (0.02)	0.86 (0.02)	0.89 (0.02)
3	10	3	Rank	1.99	0.82 (0.02)	0.79 (0.03)	0.62 (0.03)	0.82 (0.02)	0.81 (0.03)	0.88 (0.02)
4	10	4	Triweight	1.97	0.79 (0.02)	0.87 (0.03)	0.66 (0.03)	0.80 (0.02)	0.87 (0.03)	0.88 (0.02)

PR-AUC = precision recall-area under the curve.

**Table 6 diagnostics-12-02826-t006:** Performance metrics across mammographic density on the validation dataset.

Performance Metrics	Validation Dataset		
	Overall	Non-Dense	Dense
Sensitivity	0.74	0.76	0.71
Specificity	0.34	0.25	0.43
Youden J index	0.08	0.01	0.15
F2 score	0.75	0.77	0.73
Precision	0.80	0.80	0.81
PR-AUC	0.82	0.83	0.82

## Data Availability

The data are available upon reasonable request to the corresponding author.

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
