# Peer review of "Over-the-Counter Breast Cancer Classification Using Machine Learning and Patient Registration Records"

_diagnostics, 2022, doi:10.3390/diagnostics12112826_

Round 1

Reviewer 1 Report

The article submitted to me for review covered the very emerging topic for society with a worldwide influence.

The following comments should be addressed from the authors:

1. The following paragraph: 

Artificial Intelligence (AI) is a subfield of computer science that aims to develop a system capable of performing a task that usually requires human intelligence. The rise of AI is expected to improve many areas including the fields of healthcare and medicine. AI had been studied to be used as a medical analytic tool including for drug discovery, ge-
nomic medicine, disease prognosis and diagnosis, and personalised healthcare
[16,17]. However, the adoption of AI in healthcare and medicine is slower than in other fields [18].
Explainable AI (XAI) aims to make the AI more interpretable and understandable to the
6end-users. Thus, the use of XAI will further help in the successful implementation of AI in healthcare. Generally, the approaches used in the XAI can be divided into model-specific and model-agnostic approaches [19].

The focus of the paper is ML, then please omit and extend the Intro part with cases that were treated with ML. It will be good to comment on the capability of ML for other diseases such as diabetis, for example.

Next, what was the driving force for choosing the following spliting:

The data was split into 90% development dataset and 10% validation dataset.

Did you try 80 to 20?

Do the authors used a target set?

3.3. Explainable machine learning- not clear, change the title

KNN is a sensitive method to the size of thr dataset, the dataset was not too large, then how you can prove that Knn was the best? Please, add more model validations metrics.

Was not clear if the authors are using the transform or real data? 

Figure 5 illustrates the top fifteen influential features of the final ML model. - PCA should be used to prove the stated selection.

kNN model had a slightly higher sensitivity - such a statment can't convinse me that Knn was the best option.

What was the SVM modrl parameters? Add the explanation about which types of SVM was used for the data-driven model?

Reviewer 2 Report

1.   Use word Prevention or Classification instead of screening in the title to match the hierarchy in the paper.

2.   The abstract should be redefined, it is suggested to give numerical result of the carried-out work, also the keywords are too less it should be 6 to 10, avoid using abbreviations in the keywords. 

3.   Introduction section looks week authors should give brief introduction of the carried-out work. Authors must define the contribution about the work, motivation behind the work, and problem statement as separate subsection of the introduction section.

4.   Limit the references to 10 in the introduction section and remove all the unnecessary references in the entire manuscript.

5.   Section 2 should be related work section of the existing state of the art literature and comparison table among all the existing techniques.

6.   Write the proposed section after the material and methods section to improve the quality and novelty of the manuscript.

7.     All the tables, figures, and equations must be referred in text and before it should be drawn in the text.  

8.   Table 3 should be complete in all the aspects.

9.   Figure 4 and figure 5 should be improve in terms of quality and clarity.

10.  Do not cite the manuscript in this way like [5,70–72].

11.  Conclusion could be minimum of 300 words with detailed analysis of results.

12.  Authors have not carried out the extensive literature survey some new recent works required to be added, a few links are given below for consideration

https://www.sciencedirect.com/science/article/pii/S1879406822001965

https://www.mdpi.com/2075-4418/11/2/241

https://www.sciencedirect.com/science/article/pii/S266635112200047X

https://www.sciencedirect.com/science/article/pii/B9780128181485000151

https://www.sciencedirect.com/science/article/pii/S0010482520301463

Round 2

Reviewer 1 Report

The authors addressed all my queries. Thank  you for your detailed response.

Reviewer 2 Report

Authors revised the paper according to the suggestions.